# Assessing vulnerability for future Zika virus outbreaks using seroprevalence data and environmental suitability maps

Yannik Roell[1], Laura Pezzi[2,3], Anyela Lozano-Parra[4], Daniel Olson[1,5], Jane Messina[6,7], Talia Quandelacy[8], Jan Felix Drexler[9], Oliver Brady[10,11], Morteza Karimzadeh[12], Thomas Jaenisch[1,13]*

1 Center for Global Health, Colorado School of Public Health, University of Colorado, Aurora, Colorado, United States of America, 2 National Reference Center for Arboviruses, Inserm-IRBA, Marseille, France, 3 Unité des Virus Émergents (UVE: Aix-Marseille Univ, Universitá di Corsica, IRD 190, Inserm 1207, IRBA), France, 4 Grupo de Epidemiología Clínica, Universidad Industrial de Santander, Bucaramanga, Colombia, 5 Division of Pediatric Infectious Diseases, University of Colorado School of Medicine, Aurora, Colorado, United States of America, 6 School of Geography and the Environment, University of Oxford, Oxford, United Kingdom, 7 Oxford School of Global and Area Studies, University of Oxford, Oxford, United Kingdom, 8 Department of Epidemiology, University of Colorado, Aurora, Colorado, United States of America, 9 Institute of Virology, Charité-Universitätsmedizin Berlin, corporate member of Freie Universität Berlin and Humboldt Universität zu Berlin, Berlin, Germany, 10 Department of Infectious Disease Epidemiology, Faculty of Epidemiology and Population Health, London School of Hygiene & Tropical Medicine, London, United Kingdom, 11 Centre for Mathematical Modelling of Infectious Diseases, Faculty of Epidemiology and Population Health, London School of Hygiene & Tropical Medicine, London, United Kingdom, 12 Department of Geography, University of Colorado, Boulder, Colorado, United States of America, 13 Heidelberg Institute of Global Health (HIGH), Heidelberg University Hospital, Heidelberg, Germany

* thomas.jaenisch@cuanschutz.edu

**Data Availability Statement:** All relevant data are in the manuscript and its supporting information files.

## Abstract

The 2015–17 Zika virus (ZIKV) epidemic in the Americas subsided faster than expected and evolving population immunity was postulated to be the main reason. Herd immunization is suggested to occur around 60–70% seroprevalence, depending on demographic density and climate suitability. However, herd immunity was only documented for a few cities in South America, meaning a substantial portion of the population might still be vulnerable to a future Zika virus outbreak. The aim of our study was to determine the vulnerability of populations to ZIKV by comparing the environmental suitability of ZIKV transmission to the observed seroprevalence, based on published studies. Using a systematic search, we collected seroprevalence and geospatial data for 119 unique locations from 37 studies. Extracting the environmental suitability at each location and converting to a hypothetical expected seroprevalence, we were able to determine the discrepancy between observed and expected. This discrepancy is an indicator of vulnerability and divided into three categories: high risk, low risk, and very low risk. The vulnerability was used to evaluate the level of risk that each location still has for a ZIKV outbreak to occur. Of the 119 unique locations, 69 locations (58%) fell within the high risk category, 47 locations (39%) fell within the low risk category, and 3 locations (3%) fell within the very low risk category. The considerable heterogeneity between environmental suitability and seroprevalence potentially leaves a large population vulnerable to future infection. Vulnerability seems to be especially pronounced at the fringes of the environmental suitability for ZIKV (e.g. Sao Paulo, Brazil). The

**Funding:** The author(s) received no specific funding for this work.

**Competing interests:** The authors have declared that no competing interests exist.

discrepancies between observed and expected seroprevalence raise the question: "why did the ZIKV epidemic stop with large populations unaffected?". This lack of understanding also highlights that future ZIKV outbreaks currently cannot be predicted with confidence.

## Author summary

After the ZIKV epidemic in the Americas, it remains unclear if and when a resurgence of the ZIKV could occur. We used publicly available data on environmental suitability of transmission as well as seroprevalence data to estimate future vulnerability to ZIKV outbreaks. Our results show a considerable discrepancy between the observed seroprevalence, from past exposure to the virus on one hand, and the environmental suitability, which raised the question why the epidemic subsided before reaching the expected herd immunization threshold in many locations. This lack of understanding also highlights that future ZIKV outbreaks currently cannot be predicted with confidence. Although we cannot provide an answer to the question why the epidemic subsided when it did, we present a better quantification and geospatial mapping of the potential vulnerability to future outbreaks, which will be crucial for decision makers to prepare for future outbreaks.

## Introduction

Zika virus (ZIKV) is an arthropod-borne virus (arbovirus) belonging to the *Flaviviridae* family of the *Flavivirus* genus. The virus is transmitted to humans through the bite of infected mosquitoes from the *Aedes* (*Ae.*) genus, mainly *Ae. aegypti* and *Ae. albopictus*. Non-vectorial transmission routes have also been described, including sexual transmission, mother-to-child transmission (vertical transmission), and blood transfusion [1].

Two geographically distinct lineages of ZIKV have been identified so far: African and Asian. The African-ZIKV lineage likely originated in Eastern Africa and then spread west, causing sporadic or recurrent infections in African countries [2]. The first Asian ZIKV was isolated in Malaysia in the late 1960s and then the virus spread across Southeast Asia and the Pacific area, where the virus was responsible for localized outbreaks in Micronesia [3] and French Polynesia [4]. In 2015, the Asian lineage caused an explosive epidemic in the Americas; and the World Health Organization (WHO) declared ZIKV a Public Health Emergency of International Concern (PHEIC) in February 2016 [5]. Because of the similarities with other arboviral infections, Zika infections were first misdiagnosed as dengue at the beginning of the epidemic [5,6]. In 2017, ZIKV transmission in the Americas dropped sharply.

Several ZIKV seroprevalence studies have been performed in the Americas, Asia, and Africa; however, these estimates are affected by the strong cross-reactivity between anti-ZIKV antibodies and antibodies against other Flaviviruses, especially dengue virus (DENV) that is endemic in large areas of these continents where it co-circulates with ZIKV [7]. Neutralization tests are the serology gold standard because of their capacity to differentiate ZIKV infections from other closely-related viruses, while rapid immunochromatographic tests (RDT) and enzyme-linked immunosorbent assay (ELISA) have proved to be generally characterized by low specificity [8–13].

The end of the large outbreak in the Americas was attributed to evolving population immunity [14]. The expected herd immunity threshold for Zika virus was estimated at around 65% [3,13]. Seroprevalence estimates for the bigger cities along the Atlantic coast in Brazil

confirmed this estimate [13,15], but smaller cities more inland in Brazil seem not to have been affected to the same degree [16]. Because the rapid end of the ZIKV epidemic in the Americas was poorly understood, we aimed to evaluate potential vulnerability for future epidemics. Here we analyze seroprevalence estimates and correlate these with environmental suitability for ZIKV transmission. Our hypothesis is that areas with high environmental suitability and comparably lower seroprevalence estimates may be at future risk for ZIKV emergence.

## Methods

We plotted (a) published seroprevalence results from the literature (observed seroprevalence) and compared these with (b) environmental suitability maps. The difference between the environmental suitability and observed seroprevalence was used to determine (c) the vulnerability of a location having a severe outbreak if another epidemic occurred. The risk categories (high, low, or very low risk) were combined with perceived covariates to (d) determine if any differences between the risk categories exist.

### Literature search for seroprevalence estimates

We retrieved geospatial seroprevalence data for Zika virus from PubMed and LILACS on October 22, 2021. Using "seroprevalence zika" as the search term, 139 records were returned. Each of the abstracts were reviewed by one person to exclude papers that did not meet our criteria. Afterwards, one reviewer screened the full text of the remaining papers, and a second reviewer was used to confirm papers that were excluded. We found 37 eligible studies that included point specific ZIKV seroprevalence (Fig 1). An eligible study included point location seroprevalence values for humans, the geographic region was either in the Americas, Africa, or Asia/Oceania, and participants were not symptomatic or febrile patients. For studies without geolocation information reported, we reached out to the authors via email up to three times before eliminating that record. Locations in Oceania are binned into the Asia group in the remaining paper due to proxy. After removing populations that were symptomatic or febrile patients, each location from a study was divided into one of four populations: pregnant women, blood donors, community, and other. From the 37 studies, 119 unique locations were

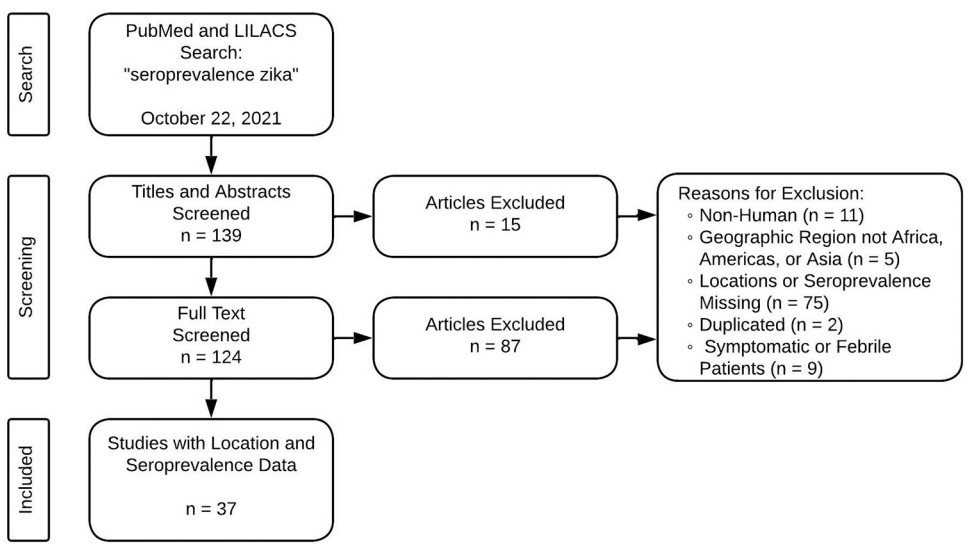

**Fig 1. Flow chart of literature search and screening results.**

identified across 26 different countries (Fig 2). If more than one seroprevalence estimate was available from a single location (e.g. sequential surveys), we kept the latest and highest value.

The serology assays used were grouped into two categories, according to their performances in terms of sensitivity and specificity: (1) neutralization tests (NT) and (2) all other tests which includes enzyme-linked immunosorbent assay (ELISA), rapid diagnostic test (RDT), microsphere immunoassay (MIA), and hemagglutination inhibition assay (HIA). These categories were chosen because NTs are considered serology gold standards and the other tests are strongly affected by cross-reactivity with antibodies against other Flaviviruses, lacking specificity, or they lack sensitivity.

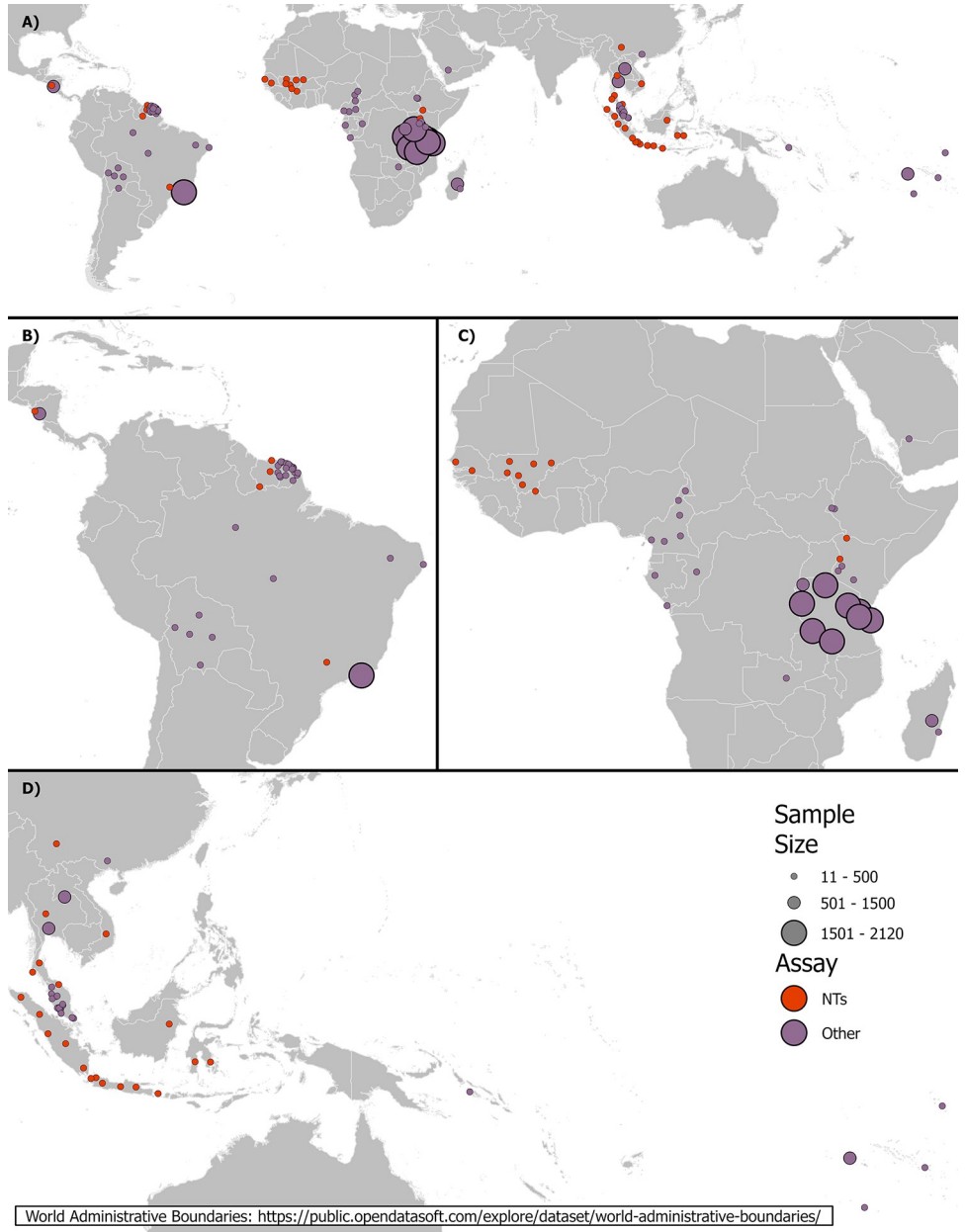

**Fig 2. (A) Global study locations with sample size and serological assay group (NTs–neutralization tests and Other) and showing greater detail for (B) the Americas, (C) Africa, and (D) Asia. The world administrative boundary basemap is from** https://public.opendatasoft.com/explore/dataset/world-administrative-boundaries/.

## Environmental suitability maps

For environmental suitability, or ZIKV transmission risk, we used maps published by Messina, Kraemer [17]. While technically producing predictions of probability of one or more cases, these suitability maps have been previously shown to correlate with incidence [18]. The methodology is described in detail in the publication, but as a summary, these maps were made from ensemble boosted regression with ZIKV occurrence data in humans and absence background locations. Six covariates thought to influence ZIKV transmission were used to determine the relationship between presence or absence and the environmental conditions. The covariates are temperature suitability for dengue transmission to humans via *Ae. aegypti* and *Ae. albopictus*, minimum relative humidity, annual cumulative precipitation, enhanced vegetation index, and urban versus rural habitat type. At the time of the published environmental suitability map, many of the parameters the authors used were not measured experimentally for ZIKV but for the closely related DENV, which is why their covariates are for dengue transmission. The generated 5-km resolution environmental suitability map ranges from 0 to 1 with 1 being the most suitable.

## Vulnerability estimation and mapping

To assess populations vulnerable to a ZIKV outbreak, we estimated the discrepancy between the observed seroprevalence and predicted ZIKV environmental suitability:

*Vulnerability = Seroprevalence (from studies)–Environmental Suitability (from map)* (1)

Areas with higher environmental suitability would be expected to have higher seroprevalence. While we understand that environmental suitability and expected seroprevalence is not a one-to-one relationship due to the herd immunity being reached around 60–70% seroprevalence [3], this is being used as a proxy to understand the differences between what is observed and expected. We believe areas that are vulnerable will be places where the environmental suitability is higher than the seroprevalence (negative vulnerability). Vulnerability has been divided into three categories (high, low, and very low risk) depending on the distribution of the data.

## Exploratory data analysis

We examined the relationship between vulnerability and temperature (mean temperature in coldest quarter) [19], human population density [20], and socioeconomic status [21] using Spearman's and Pearson's correlation coefficients, as well as by visual inspection (Table 1). Socioeconomic status was chosen because people with lower socioeconomic status often live in areas with uncontrolled urbanization and deteriorating housing, water, and waste management systems which create ideal conditions for transmission of mosquito-borne disease [22,23]. All maps were resampled to have a resolution of 5 km and the values from the maps

**Table 1. Description of maps utilized within the exploratory data analysis of clusters.**

| Variable | Global Range | Description | Source |
|---|---|---|---|
| Environmental Suitability | 0–1 | Environmental suitability for Zika virus with higher values indicating more suitable | Messina, Kraemer [17] |
| Temperature | -66–29˚C | Mean temperature of coldest quarter | Fick and Hijmans [19] |
| Human Population Density | 0–103341 ppl / km$^2$ | Estimated total number of people per cell | Doxsey-Whitfield, MacManus [20] |
| Socioeconomic Status | 0–1020 | Purchasing power parity | Nordhaus and Chen [21] |

were extracted at each study location using ArcGIS Pro 2.8 [24]. If a point was not within a raster cell, the nearest value was used, except for points within French Guiana. Due to the socio-economic status layer not depicting a value for French Guiana, we used the lowest value available for mainland France as the default value for all points in French Guiana. Finally, the Mann Whitney U test was implemented to determine if there are differences that can be detected between the high risk and low risk categories for the three predictor variables.

## Results

### Data selection

The data from the 119 locations cover very low (0%, seven locations with a sample size ranging from 11 to 366) to high seroprevalence (62%, sample size = 105), with studies ranging in sample size from 11 to 2120 individuals (Fig 2). The most common of the four distinct populations assigned to each location was community (N = 75), with pregnant women being the least (N = 6). Blood donors were designated in 20 locations and a total of 18 locations assigned as other. The average seroprevalence for ZIKV was 16%. Africa had the lowest average seroprevalence (8%, SD = 8%), Asia's average seroprevalence was 15% (SD = 13%), and the Americas showed the highest average seroprevalence (24%, SD = 18%). No location in Africa had a higher seroprevalence than 50% while two locations in Asia and six locations in the Americas scored above 50%. Only 36 locations used a gold standard NT assay while 83 locations used an assay assigned in the other group.

All three continents were represented about equally with the number of locations (Africa N = 37, Americas N = 38, and Asia N = 44), with dates of sample collection ranging from July 2010 to June 2019 in Africa, December 2015 to December 2018 in the Americas, and December 2012 to October 2019 in Asia. Africa had the most countries represented with 12 countries, Asia had 9 countries represented, and the Americas had 5 countries represented. The most locations in one country were in French Guiana (N = 22) with the remaining countries in the Americas ranging between 2–6. Malaysia had 15 locations and Indonesia had 14 locations while the remaining countries in Asia ranged between 1–5 locations. Africa was evenly distributed with 1–8 locations per country. Detailed information for each study can be found in S1 Table.

### Vulnerability

The three classes of vulnerability (high, low, and very low risk) were chosen based on the distribution of environmental suitability vs seroprevalence data (Fig 3). A value of 0.6 was used as the cutoff to distinguish between the areas with higher vs lower environmental suitability and a value of 0.5 was used as the cutoff to distinguish between the areas with higher vs lower seroprevalence. The cutoff values were chosen based on the division seen when visualizing the data. Locations that have an environmental suitability value greater than 0.6 and a seroprevalence less than 0.5 are indicated as high-risk population. A total of 69 locations (58%) fell within the high risk category. Locations with an environmental suitability value greater than 0.6 and a seroprevalence greater than 0.5 or an environmental suitability value less than 0.6 and a seroprevalence less than 0.5 were considered low risk, although these two different sub-categories have different underlying risk profiles. A total of 47 locations (39%) fell within this low risk category. Locations with an environmental suitability value less than 0.6 and a seroprevalence greater than 0.5 were evaluated as being very low risk. A total of 3 locations (3%) fell within the very low risk category. The difference between the low risk categories is that the very low risk has a low transmission risk with high seroprevalence, resulting in an outbreak unlikely to develop into a major outbreak due to existing herd immunity. The low risk either

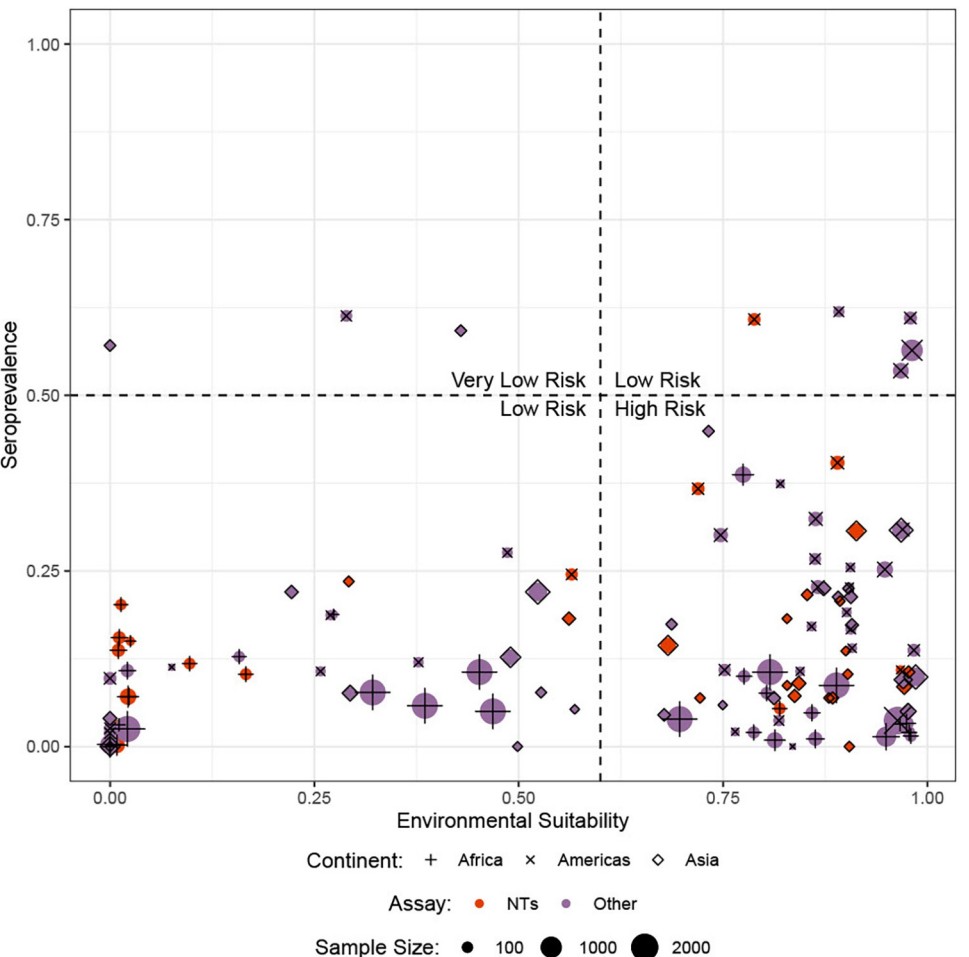

**Fig 3. Division of vulnerability based on the difference between seroprevalence and environmental suitability. NTs: neutralization tests.**

has a low transmission risk, resulting in the risk of a small outbreak if transmission is established, or has a high transmission risk but also has a high seroprevalence, resulting in a major outbreak not likely occurring due to herd immunity. The distribution of vulnerability categories along with environmental suitability can be found in Fig 4.

## Exploratory data analysis

We investigated temperature, population density, and socioeconomic status as candidate explanatory variables for vulnerability. Using Spearman's and Pearson's correlation, all variables were statistically significant (p-value < 0.5) with at least one of the correlation metrics and negatively correlated with vulnerability. Temperature and vulnerability were significantly correlated using Pearson's correlation ($r$ = -0.25, p-value = 0.006), as well as population density (Pearson: $r$ = -0.25, p-value = 0.006). Socioeconomic status was significantly correlated with vulnerability using both methods (Spearman: $r$ = -0.34, p-value = 0.0001; Pearson: $r$ = -0.21, p-value = 0.03).

The data was divided into two groups for the Mann Whitney U test: high risk points as one group and low risk points having a seroprevalence below 0.5 as another group. The two groups

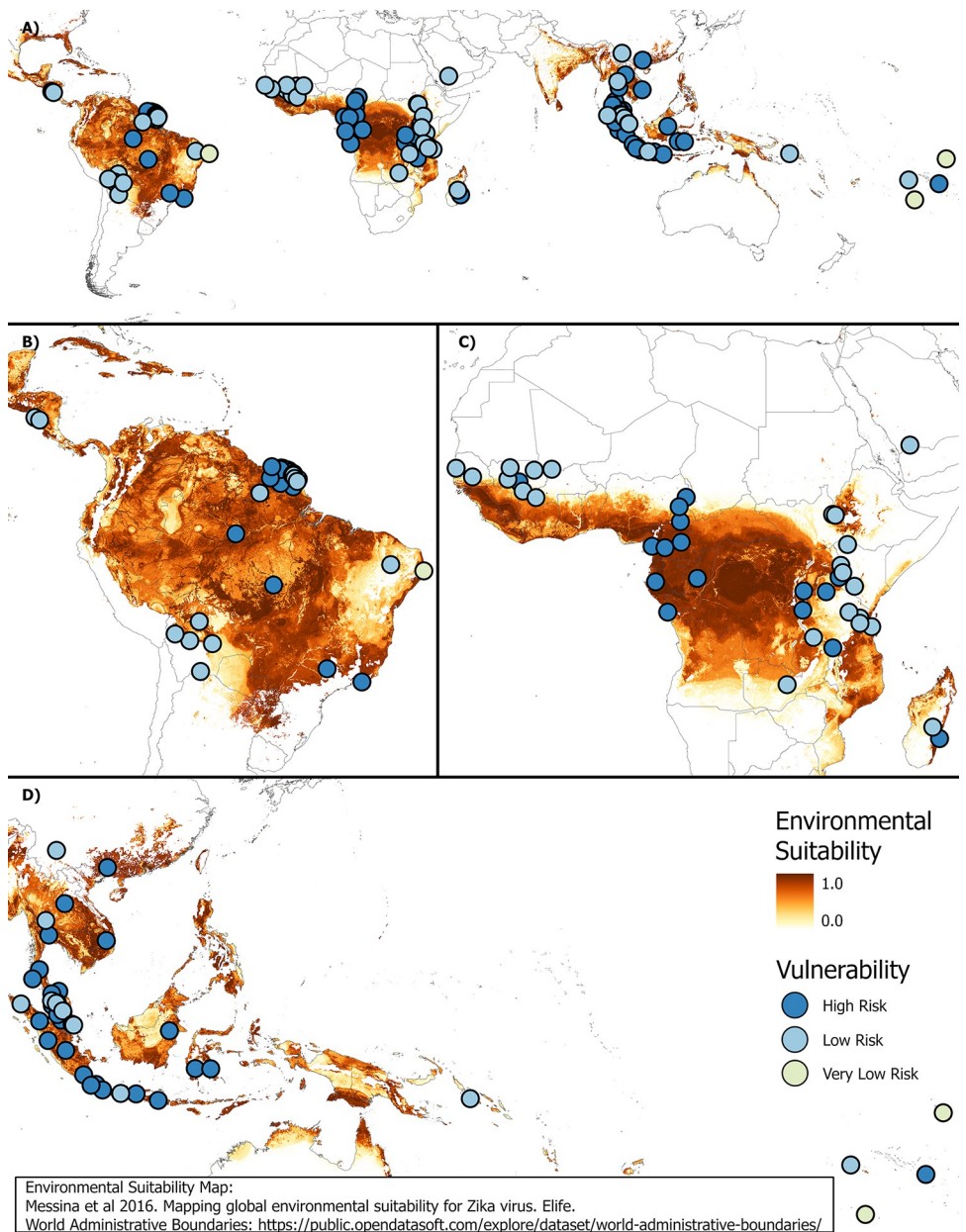

**Fig 4. (A) Global distribution of vulnerability for ZIKV with environmental suitability depicted and showing greater detail for (B) the Americas, (C) Africa, and (D) Asia. The environmental suitability map is from Messina, Kraemer [17] and the world administrative boundary basemap is from** https://public.opendatasoft.com/explore/dataset/world-administrative-boundaries/.

were significantly different regarding temperature (p-value = 0.004) and socioeconomic status (p-value = 0.01). For population density, the two groups did not show a statistically significant difference (p-value > 0.05) using the Mann Whitney U test.

## Discussion

Here we present an assessment of potential vulnerability for future ZIKV transmission and subsequent outbreaks based on seroprevalence and environmental suitability. ZIKV

resurgence remains a real threat for Latin America where many regions seem to have reached lower seroprevalence during the ZIKV epidemic than what would be considered protective on a population level, despite environmental suitability. The herd immunization threshold for ZIKV was estimated at around 65% based on data from Yap Island, Micronesia [3] and Salvador de Bahia, Brazil [13]. The reason why the ZIKV epidemic rapidly subsided in Latin America in 2016–17 remains poorly understood, in spite of the fact that models based on incidence trends (not seroprevalence) were able to predict that most major cities in Latin America had experienced a transmission peak before 2018 and would not experience high case numbers in 2018 and onwards [14]. In some areas, co-circulation of other arboviruses such as chikungunya virus (CHIKV; hardly affected South America between 2015 and 2017) and DENV (with the four serotypes hyper-endemic in many Latin American countries) might have played a role in replacing ZIKV transmission [25]. Since these arboviruses share the same hosts and vectors, several factors such as viral competition in both humans and mosquitoes, or existing population immunity against these pathogens, could have influenced ZIKV transmission patterns.

For Africa, the assessment of vulnerability is more uncertain, even if we observed regions with lower seroprevalence estimates despite high environmental suitability for ZIKV transmission. Population immunity to ZIKV in Africa is likely affected by ongoing transmission of other related flaviviruses. In the past, ZIKV outbreaks and clusters of microcephaly or Guillain Barré Syndrome were not reported–but the surveillance systems in Africa might also not be equipped to detect a signal unless very prominent.

## Limitations of the study

The relationship between environmental suitability and transmission probability is likely non-linear. For our purposes, we have simplified this relationship to be linear. We also acknowledge that uncertainties regarding the mapping scale and location data can lead to misinterpretations, especially since the suitability map was produced early in the Zika epidemic. The underlying environmental suitability map lacks granularity, which has the potential to produce questionable results, at least in the small scale, which means the data needs to be interpreted with caution and we therefore recommend interpreting the results as an approximation of the true risk and the true location.

Our data is based on different populations (pregnant women, blood donors, community, and other special populations) whose representativity cannot be validated, as well as on heterogenous testing strategies and assays, which are shown in Fig 3. We acknowledge that different assay performance has an influence on the precision of the seroprevalence estimates, which we cannot adjust for in this analysis. This influence would likely not alter the overall interpretation of the remaining pockets of vulnerable populations.

We assumed that ZIKV circulation induces protective immunity against both homologous and heterologous ZIKV lineages, since cross-neutralization has been observed using both human and murine immune sera [26]. However, higher titers are always observed with the homologous strain and the introduction of heterologous ZIKV lineage in an area where it has never circulated before (e.g. African lineage imported into America) can possibly have an impact on population susceptibility. Also, a decline in ZIKV neutralizing antibodies over time after natural infection has been observed, underlying that previous ZIKV circulation is not synonymous of protection [27].

## Conclusion

Our results show a potential vulnerability for increased future ZIKV transmission in many areas of Latin America, as well as in Africa and Asia where anti-ZIKV seroprevalence rates are

low and environmental suitability for transmission is high. Because of the difference in the background epidemiology of ZIKV and related flaviviruses in Asia and Africa, this finding needs to be interpreted cautiously outside of Latin America.

The recent resurgence of other arboviruses in Latin America suggests that Zika re-emergence might also be a risk in areas where population immunity was not reached or has decreased. Acknowledging the inherent limitations, seroprevalence surveys can be a useful tool to assess the population vulnerability to future outbreaks.

## Supporting information

**S1 Table. The dataset generated from extracting data from seroprevalence studies and values from associated maps (minimal data set for analysis).**
(XLSX)

## Author Contributions

**Conceptualization:** Yannik Roell, Thomas Jaenisch.

**Data curation:** Yannik Roell, Thomas Jaenisch.

**Formal analysis:** Yannik Roell.

**Funding acquisition:** Jan Felix Drexler, Thomas Jaenisch.

**Investigation:** Yannik Roell.

**Methodology:** Yannik Roell, Laura Pezzi, Talia Quandelacy.

**Project administration:** Yannik Roell.

**Resources:** Anyela Lozano-Parra.

**Software:** Yannik Roell.

**Supervision:** Oliver Brady, Morteza Karimzadeh, Thomas Jaenisch.

**Validation:** Yannik Roell, Laura Pezzi, Anyela Lozano-Parra, Daniel Olson, Jane Messina, Talia Quandelacy, Jan Felix Drexler, Thomas Jaenisch.

**Visualization:** Yannik Roell, Jane Messina, Oliver Brady, Morteza Karimzadeh.

**Writing – original draft:** Yannik Roell, Laura Pezzi, Daniel Olson, Thomas Jaenisch.

**Writing – review & editing:** Yannik Roell, Laura Pezzi, Anyela Lozano-Parra, Daniel Olson, Jane Messina, Talia Quandelacy, Jan Felix Drexler, Oliver Brady, Morteza Karimzadeh, Thomas Jaenisch.

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
