## [Decision Letter · Decision Letter 0]

21 Sep 2023

Dear Dr. Jaenisch,

Thank you very much for submitting your manuscript "Assessing vulnerability for future Zika virus outbreaks using seroprevalence data and environmental suitability maps" for consideration at PLOS Neglected Tropical Diseases. As with all papers reviewed by the journal, your manuscript was reviewed by members of the editorial board and by several independent reviewers. In light of the reviews (below this email), we would like to invite the resubmission of a significantly-revised version that takes into account the reviewers' comments. 

Please revise the manuscript to incorporate the comments of the three reviewers. In particular, include a justification for restricting the scope of the analysis to Africa and the Americas to the exclusion of Asia, and for limiting the literature database search to PubMed.

We cannot make any decision about publication until we have seen the revised manuscript and your response to the reviewers' comments. Your revised manuscript is also likely to be sent to reviewers for further evaluation.

Sincerely,

Josh M Colston, Ph.D.

Guest Editor

Andrea Marzi

Section Editor

Please revise the manuscript to incorporate the comments of the three reviewers. In particular, include a justification for restricting the scope of the analysis to Africa and the Americas to the exclusion of Asia, and for limiting the literature database search to PubMed.

Reviewer's Responses to Questions

**Key Review Criteria Required for Acceptance?**

**Methods**

-Are the objectives of the study clearly articulated with a clear testable hypothesis stated?

-Is the study design appropriate to address the stated objectives?

-Is the population clearly described and appropriate for the hypothesis being tested?

-Is the sample size sufficient to ensure adequate power to address the hypothesis being tested?

-Were correct statistical analysis used to support conclusions?

-Are there concerns about ethical or regulatory requirements being met?

Reviewer #1: See below

Reviewer #2: (No Response)

Reviewer #3: The text clearly articulates the objective of the study, which is to test the hypothesis that regions that areas with high environmental suitability and comparably lower seroprevalence estimates may be at future risk for ZIKV emergence.

To test the hypothesis that regions characterized by high environmental suitability and relatively lower seroprevalence estimates may face future risks related to ZIKV emergence, it is essential to consider a broader geographical scope. Excluding regions such as Southeast Asia from the analysis could limit the applicability of the findings and hinder a comprehensive understanding of the global Zika virus risk landscape. Given the uncertainty surrounding Zika virus evolution, providing a rationale for the exclusion of Southeast Asia becomes crucial. While Southeast Asian populations are thought to exhibit some resistance to ZIKV infection, the ongoing trends of globalization and international travel create a significant risk for the potential establishment of endemic ZIKV circulation in this region. This underscores the importance of justifying the decision to exclude Southeast Asia from the analysis.

In the context of conducting a literature search for seroprevalence estimates, it is acknowledged that PubMed stands as the suitable database for bioscience-related inquiries. However, it is important to address the question of why PubMed was not supplemented with other databases to enhance the comprehensiveness of information retrieval.

The furnished information delineates the methodology employed in the research to evaluate the correlation among observed seroprevalence, environmental suitability maps, vulnerability, and risk categories. Nevertheless, with regard to the adequacy of the sample size for ensuring adequate statistical power in addressing the tested hypothesis, augmenting the dataset from other sources would have augmented the information.

The statistical analyses employed to support the conclusions are considered acceptable.

No ethical concerns.

**Results**

-Does the analysis presented match the analysis plan?

-Are the results clearly and completely presented?

-Are the figures (Tables, Images) of sufficient quality for clarity?

Reviewer #1: See below

Reviewer #2: (No Response)

Reviewer #3: - In lines: 131 and 132: “Most of the locations for Africa were in French Guiana (N = 22) with the other 15 countries having between 1-8 locations represented.” French Guiana actually is in South America.

- Line 141. Defining vulnerability solely based on seroprevalence lower than environmental suitability leads to oversimplification of the results. This approach fails to identify other potential forms of vulnerability. In Figure 4, the Peruvian coastline does not exhibit environmental suitability, but it does not reflect vulnerability either. This is an error, as the presence of the vector mosquito, socio-economic conditions, temperature, and other factors are relevant in this area. Similarly, the state of Pernambuco experienced a high incidence of cases and possesses all the risk-inducing conditions. However, according to statistical analyses, it is depicted as a very low-risk area. This discrepancy raises questions about the effectiveness of the current assessment.

- Assuming a linear relationship may lead to inaccurate risk assessments, as the actual relationship between environmental suitability and transmission is likely more complex, potentially resulting in misallocation of resources for disease control.

-Table 1- Temperature -66-29C, is Mean temperature of coldest quarter. Source Fick and Hijmans (2017) correct?

**Conclusions**

-Are the conclusions supported by the data presented?

-Are the limitations of analysis clearly described?

-Do the authors discuss how these data can be helpful to advance our understanding of the topic under study?

-Is public health relevance addressed?

Reviewer #1: See below

Reviewer #2: (No Response)

Reviewer #3: The limitations of analysis are clearly described. However, it fails in incorporate in the discussion, within the limitations of the review, the role of asymptomatic individuals in assessing vulnerability for potential Zika outbreaks in the future.

The rationale behind the omission of data from Asia, despite the prevailing uncertainty concerning the trajectory of the Zika Virus (ZIKV), should be elaborated upon. Incorporating data from Asia into the analysis is highly advisable as it aims to demonstrate that regions characterized by substantial environmental suitability and relatively lower seroprevalence estimates may potentially face future risks pertaining to the emergence of the Zika virus.

**Editorial and Data Presentation Modifications?**

Reviewer #1: See below

Reviewer #2: (No Response)

Reviewer #3: N/A

**Summary and General Comments**

Reviewer #1: The manuscript submitted by Roell et al aims at examining estimating vulnerability to future Zika virus outbreaks. The authors make use of seroprevalence data and environmental suitability maps previously published. 

It is a fascinating aim, but the study has major pitfalls. The biggest one being the serology data used – with no control of the representativeness of the studies included. The statistical analysis also lack substance.

Main comments: 

- Seroprevalence data: the authors used a systematic search to retrieve geospatial seroprevalence data. A major criteria that is lacking in the selection process is the representativeness of the studied population: while the authors make sure the seroprevalence studies were done on human populations, it is critical to also control for the population considered. For example, the study with the highest seroprevalence (94%, Collins et al 2020) was done on a cohort of pregnant women. This might not be representative of the whole population. Another example: the study of Oderinde et al 2020 (prevalence of 75%) analyzed samples “from patients exhibiting febrile illness” – therefore the study is not describing the seroprevalence in the general population, but rather among people with Zika-like symptoms. This study should be removed. I suggest authors should carefully review the studies they include. When there is a doubt about the representativeness of the population (e.g. pregnant women), authors should discuss the impact on their analysis. 

- Type of assay: Authors point out that different assays are used to produce estimates of seroprevalence, but don’t really discuss the difference in estimates, NT assays consistently give seroprevalence <25%, could the authors discuss this? It makes 100% of the corresponding locations in the ‘Low Risk’ category.

- Date of seroprevalence studies: authors set out to assess the ‘vulnerability to future ZIKV outbreaks’ – however the timing of the seroprevalence studies is not taken into account, even when multiple studies are available for the same country. Can the authors be more precise on their definition of future?

- Categories: the cutoff at 45% is not as clear to me as the authors state, why not 50%?

- Clustering: Authors chose 3 different risk categories - which makes 4 different areas on figures 3 and 5. Why did the authors perform a k-mean clustering using only 3 centroids? By definition it cannot match their handmade categories.

Minor comments:

- Line 131: French Guiana is in America, not Africa

- Line 179: “None of the explanatory variables were highly correlated with vulnerability” – define “highly”. Temperature is still statistically significantly correlated.

- Lines 220-222: “available seroprevalence data may be under-reported due to the high frequency of non-symptomatic infections and that the data are based on clinically apparent infections (Duong et al., 2017).” – do you mean epidemiological data?

Reviewer #2: This is an interesting study analysing post-epidemic seroprevalence estimates in the Americas and Africa and investigating their relationship with environments suitable for ZIKV infection. Firstly, I would like to congratulate the authors for piecing together and evaluating an extensive data set. Overall, I very much enjoyed reading this manuscript. The manuscript is well written, proper statistical methods of analysis have been carried out and well explained. I do have a few comments that I hope will help improving the manuscript, or maybe just give the authors food for thoughts.

The main problems I have with the current manuscript relate to review methodology and statistical analysis. Firstly, the authors used PubMed to examine 135 records matching the term “seroprevalence zika”, but I think it would be more appropriate to define a broader range of literature databases and search terms to determine a more appropriate search strategy. An example would be to include PubMed/MEDLINE, Web of Science, and Scopus in the scope, define several proximate terms other than “seroprevalence zika”, improve the flowchart and thus perhaps give a different perspective. In doing so, it is also felt that it would be helpful to include a table of information on the complete literature used in each database and the complete search syntax as a thinning document. I also feel that clarification is needed as to whether the reviewers independently screened the titles and abstracts, or if there are discrepancies in the reviewers' assessments, whether detailed methodology, such as by discussion or other third-party reviewer input, is addressed. Indeed, all relevant records may be worth considering, with some guidelines defined as set out below. It was also unclear whether ethical approval had been obtained. This is but food for thoughts, as I believe the current analysis to have been done appropriately.

https://pubmed.ncbi.nlm.nih.gov/30178033/　

https://pubmed.ncbi.nlm.nih.gov/26134548/

https://www.ncbi.nlm.nih.gov/pmc/articles/PMC4915647/

A seemingly puzzling fact is also observed. The authors conceptualize vulnerability as an area of lower seroprevalence than environmental suitability and define it as three categories, with the basis for setting cut-off values being a clear division based on visual inspection of the data. In fact, it is felt that a more robust approach to accurately assess these relationships would be to use continuous variables and (model) risk assessment along more geospatial gradients, rather than assigning categories One of the main weaknesses of the K-means method is that it is designed for grouping under a mechanical algorithm that is only interested in the characteristics of the variables, and the correlation coefficient only evaluates linear relationships that cannot account for time delays, particularly in the analysis of complex indicators such as seroprevalence, may not adequately account for valid causal relationships. For example, temperature and vulnerability are described as positively correlated (r = 0.33) but are found to be very weakly correlated, population density and socio-economic status are also uncorrelated, and the high and low risk clusters do not seem to differ that much for any of the predictors, i.e. it appears very difficult to infer high environmental fitness for infection It appears to be very difficult to infer a high environmental suitability for infection.

Finally, as a comment out of curiosity, since the authors are engaged in a review of the propagation of ZIKA and an assessment of the relationship between the driving factors, how, practically speaking, is there a tangible contribution that can be made in the future? What is the significance and inherent importance of the original findings that can be drawn from this research, rather than simply a systematic scoping review or repeated exercise to fill knowledge gaps in new areas? I would not be too surprised if historical shifts in climate change, including recent extreme events, socio-economic factors and human mobility have played a major role in shaping the spatio-temporal abundances associated with ZIKA recurrence due to Vector migration and the complex direct or interaction patterns that exist between them It is not. Again, this is just food for thought.

Minor comments:

Line 71: I am not sure this statement is true; Although I feel that the low anti-ZIKV seroprevalence in many parts of Latin America and the Caribbean could be described to some extent, it may not necessarily mean that environmental suitability for infection is high. The authors have not been able to fully assess causality.

Lines 103-104: For reference, indicators other than Spearman's correlation coefficient (e.g., Pearson and Kendall) should also be evaluated to see the differences.

Reviewer #3: It is recommended that the rationale behind the exclusion of data from Asia, given the ongoing uncertainty surrounding the trajectory of the Zika Virus (ZIKV), should be thoroughly explained. The inclusion of Asian data in the analysis is strongly encouraged, as it has the potential to illustrate that regions characterized by significant environmental suitability and relatively lower seroprevalence estimates may indeed face future risks associated with the emergence of the Zika virus.

PLOS authors have the option to publish the peer review history of their article (what does this mean?). If published, this will include your full peer review and any attached files.

Reviewer #1: No

Reviewer #2: Yes: Keita Wagatsuma

Reviewer #3: No
---

## [Editor Report · Decision Letter 1]

20 Feb 2024

Dear Dr. Jaenisch,

We are pleased to inform you that your manuscript 'Assessing vulnerability for future Zika virus outbreaks using seroprevalence data and environmental suitability maps' has been provisionally accepted for publication in PLOS Neglected Tropical Diseases.

Best regards,

Josh M Colston, Ph.D.

Academic Editor

Andrea Marzi

Section Editor

Thank you for addressing all the comments thoroughly and congratulations on an excellent analysis and paper.

---

## [Editor Report · Acceptance letter]

19 Mar 2024

Dear Dr. Jaenisch,

We are delighted to inform you that your manuscript, "Assessing vulnerability for future Zika virus outbreaks using seroprevalence data and environmental suitability maps," has been formally accepted for publication in PLOS Neglected Tropical Diseases.

Best regards,

Shaden Kamhawi

co-Editor-in-Chief

Paul Brindley

co-Editor-in-Chief
